# Long Terminal Repeats of Gammaretroviruses Retain Stable Expression after Integration Retargeting

**DOI:** 10.3390/v16101518

**Published:** 2024-09-25

**Authors:** Dalibor Miklík, Martina Slavková, Dana Kučerová, Chahrazed Mekadim, Jakub Mrázek, Jiří Hejnar

**Affiliations:** 1Laboratory of Viral and Cellular Genetics, Institute of Molecular Genetics of the Czech Academy of Sciences, Vídeňská 1083, 142 20 Prague 4, Czech Republic; martina.slavkova@img.cas.cz (M.S.);; 2Laboratory of Anaerobic Microbiology, Institute of Animal Physiology and Genetics of the Czech Academy of Sciences, Vídeňská 1083, 142 20 Prague 4, Czech Republic; chahrazedbiotek@gmail.com (C.M.); kubino77@gmail.com (J.M.)

**Keywords:** retrovirus, expression, silencing, integration site, epigenetics, vectors

## Abstract

Retroviruses integrate into the genomes of infected host cells to form proviruses, a genetic platform for stable viral gene expression. Epigenetic silencing can, however, hamper proviral transcriptional activity. As gammaretroviruses (γRVs) preferentially integrate into active promoter and enhancer sites, the high transcriptional activity of γRVs can be attributed to this integration preference. In addition, long terminal repeats (LTRs) of some γRVs were shown to act as potent promoters by themselves. Here, we investigate the capacity of different γRV LTRs to drive stable expression within a non-preferred epigenomic environment in the context of diverse retroviral vectors. We demonstrate that different γRV LTRs are either rapidly silenced or remain active for long periods of time with a predominantly active proviral population under normal and retargeted integration. As an alternative to the established γRV systems, the feline leukemia virus and koala retrovirus LTRs are able to drive stable, albeit intensity-diverse, transgene expression. Overall, we show that despite the occurrence of rapid silencing events, most γRV LTRs can drive stable expression outside of their preferred chromatin landscape after retrovirus integrations.

## 1. Introduction

Retroviruses are ssRNA viruses of the *Retroviridae* family that utilize reverse transcription and subsequent genomic DNA integration in their replication cycles. After integration, the provirus becomes a permanent part of the host genome and forms a genetic platform for host transcription machinery-driven expression of genes encoded by the retroviral genome. The establishment of the permanent infection of the host cells makes retroviruses hard-to-cure pathogens as well as useful tools for gene delivery. However, proviral expression is often subject to epigenetic silencing, which induces viral latency and complicates the effective use of retroviral vectors.

The site of proviral integration is one of the factors influencing proviral expression stability [1,2,3,4,5,6]. Retroviral integration is not restricted to particular genomic positions and occurs genome-wide. Locally, the integration is affected by imperfect integrase preferences for DNA composition projected into a weak mixed motif at target sequences [7]. Globally, the integration is affected by preferences for chromatin modifications projected into skewed genera-specific genome-wide proviral integration distribution [8,9,10,11,12,13,14,15]. It is therefore often challenging to define whether a retrovirus is sensitive to the particular epigenomic environment in a given cell type or whether the observed proviral expression instability is the function of stochastic noise in proviral gene expression [16,17,18]. The question of whether and how the environment at the site of proviral integration affects proviral expression becomes important in clinics when (i) proviral expression is challenged to cure a latent reservoir of infection and (ii) stable transgene expression is required after gene delivery in cells and gene therapies.

The effect of the integration site environment on proviral expression was demonstrated in several retroviral systems. Proviruses of avian sarcoma and leukosis virus (ASLV), which are effectively silenced in mammalian cells [19], were shown to be protected from the silencing when integrated closely downstream from active promoters [3,20] or close to intragenic enhancers when the vector was strengthened by the insertion of the CpG island [20]. Human immunodeficiency virus (HIV-1) expression was demonstrated to be sensitive to integration retargeting [21], and active HIV-1 proviruses were shown to be enriched in the vicinity of enhancers [4,6,20]. Also, those silenced HIV-1 proviruses that are sensitive to latency-reversing agents tend to be integrated closer to enhancers than their reactivation-resistant (or “locked”) counterparts [4,22]. Active promoters and enhancers of the host genome were thus shown to act as features forming an environment permissive for retroviral transcription.

Gammaretroviruses (γRV), exemplified by murine leukemia virus (MLV), have conventionally been used as vectors due to their simple genomic organization, strong regulatory sequences, and high viral titers. On the other hand, the genotoxicity of gammaretroviruses in clinical trials remains a major concern. A non-random integration distribution pattern of γRVs was considered to be a major driver of the genotoxicity. γRV preintegration complex interacts through the integrase C-terminal domain with the bromodomain and extraterminal domain (BET) proteins, tethering the preintegration complex to host cell chromatin [23,24,25]. As a consequence, γRV proviruses are enriched at active gene promoters and enhancers [8,10,12,13]. The BET-integrase interaction can be disrupted by single-point mutation (W390A) in integrase, allowing the development of BET-independent (Bin) γRV vectors that are retargeted further from promoters and enhancers [26,27,28]. Fusion of the chromodomain of heterochromatin-binding protein 1β (CBX) to the integrase with W390A mutation (Bin^CBX^) further reinforced the retargeting profile of the γRV vectors. In contrast to retargeted HIV-1 proviruses, MLV Bin vector retargeting was not associated with any significant change in proviral expression and silencing. Bin self-inactivating (SIN) vectors with heterologous internal promoters displayed stable long-term expression and minimal signs of integration site selection associated with the selection of active proviruses [29].

MLV serves as a prototype virus to study the integration and expression of γRVs. Although silencing of MLV expression in murine/rat somatic cells was reported [30,31], we previously observed that the MLV-derived vector establishes long-term stable expression in the human K562 cell line [5]. Moreover, in another study, treatment of MLV-transduced HeLa cells by histone deacetylase inhibitor did not rescue any significant silenced proviral population [32]. Expression driven by MLV LTR was however identified as not sufficient in the human hematopoietic cells, and, thus, the spleen focus-forming virus (SFFV) LTR was selected as a strong γRV-derived promoter for expression in hematopoietic cells [33] and has been extensively used as an internal promoter in SIN retroviral vectors [34,35,36,37,38,39]. Indeed, SFFV LTR can drive a long-term stable expression of genes carried by MLV Bin SIN vectors [29]. However, the sensitivity of the SFFV LTR to epigenetic silencing in pluripotent stem cells [40] together with the potent transforming activity [41,42] made the SFFV LTR unsuitable for clinical applications.

The available information suggests that γRV LTRs may act as potent drivers of expression in human somatic cells. However, data about expression outside the preferred loci and activity other than MLV and SFFV-derived LTRs are scarce. In this work, we investigated the ability of distinct γRV LTRs to establish and keep stable expression in genomic sites underrepresented in normal γRV integration. We used γRV Bin vectors and alpharetroviral (αRV) SIN vectors to study the expression and silencing of γRV LTRs after modified integration targeting.

## 2. Materials and Methods

### 2.1. Cell Cultures, Viral Vector Production, and Transduction

The HEK293T cell line was cultivated in DMEM:F12 medium (Sigma-Aldrich, Darmstadt, Germany), supplemented with 5% fetal calf serum and 5% newborn calf serum (both GIBCO, Waltham, MA, USA), and antibiotics (#A5955, Antibiotic Antimycotic Solution (100×), Stabilized; Sigma-Aldrich) at 37 °C in a 5% CO_2_ atmosphere. The cell line K562 was cultivated in RPMI 1640 medium (Sigma-Aldrich), supplemented with 5% fetal calf serum and 5% newborn calf serum (both GIBCO), and antibiotics (#A5955, Antibiotic Antimycotic Solution (100×), Stabilized; Sigma-Aldrich) at 37 °C in a 5% CO_2_ atmosphere. We received the HEK293T and K562 cell lines as kind gifts from Dr. Michal Koc and Dr. Ladislav Andera. The cell lines were stored in liquid nitrogen in small aliquots, therefore the source of the cell lines matches the source used in the previous publications [5,20].

Both γRV and ASLV-derived SIN (AS).γRV vectors were produced by HEK293T cell line co-transfection. One day before the transfection, cells were seeded on polylysine-coated plates/dishes. The next day, cells were co-transfected by the viral genome, Gag-Pol [28,34,39], and pVSV-G (Clontech, Mountain View, California, USA) constructs in 6:3:1 weight ratios by X-Treme Gene HP Transfection Reagent (Roche, Basel, Switzerland) or the calcium phosphate method. The provider protocol was followed for transfection by X-Treme Transfection reagent. When the calcium phosphate method was used for transfection, 2.5–3 × 10^6^ cells were seeded on a p100 plate, and a total of 30 μg of plasmid DNA was mixed with water (up to 1 mL) and 135 μL of 2M CaCl_2_·2H_2_O, creating mix A. Mix A was then dropwise added to mix B, formed by 1.12 mL of HBS (pH 7.0–7.1) and 22 μL of 100× concentrated PO_4_. The complete transfection mix was added dropwise to the cells and washed after 5 h by HBS with glycerol and PBS, and the cells were supplemented by fresh cultivation medium. One day after transfection, the medium was changed for the fresh medium, and two days after transfection, the viral stocks were collected. Some viral stocks were concentrated using ultracentrifugation when the collected medium was first filtered through a 0.45 μm syringe filter (Corning, Corning, NY, USA) and then ultracentrifuged at 24,000 RPM using the SW40-Ti rotor for 2.5 h at 4 °C. The supernatant was then removed and the pellet was dissolved in RPMI 1640 medium (Sigma) by shaking at 4 °C overnight. The resulting viral stocks were stored at −80 °C.

K562 cells were counted and seeded with fresh cultivation medium on a cultivation plate/dish on the day of transduction. Virus-containing medium was added to the cultivation medium and cells were incubated for 24 h. The next day, the medium was removed, and fresh cultivation medium was added to the cells.

### 2.2. Plasmids and Cloning

We used pLG the MLV-derived vector with 5′ Moloney murine sarcoma virus (MoMSV) long terminal repeat (LTR) and 3′ Moloney murine leukemia virus (MoMLV) LTR expressing enhanced green fluorescent protein (EGFP) [43] for initial experiments with Bin vectors. For the experiments comparing LTR activity, the pLd2G vector was derived from the pLG vector by replacing EGFP with EGFP fused to the destabilization domain (d2GFP). The d2GFP was amplified with Phusion Hot Start DNA Polymerase (Thermo Fisher Scientific) with LdG_IF primers and inserted into EcoRI and NcoI-HF (NEB)-digested pLG plasmid by In-Fusion^®^ HD Cloning Kit (Takara, Kusatsu, Japan), resulting in the pLd2G vector. The backbone of pLd2G was amplified by gLTR_OUT_IF primers, and the d2GFP-containing genomic sequence was amplified by gLTR_IN_IF primers. The resulting gLTR_OUT and gLTR_IN amplicons were used to create the vectors with heterologous LTRs.

MoMLV LTR was amplified from pLd2G by MoMLV_LTR primers, and CrERV LTR was amplified from pCrERV-5 [44] by CrERV_LTR primers. Feline leukemia virus (FeLV, GenBank NC_001940 [45]), spleen necrosis virus (SNV, [46]), and koala retrovirus (KoRV, GenBank NC_039228 [47]) LTR sequences were synthesized as gBlocks™ Gene Fragments (Integrated DNA Technologies). LTR amplicons or gBlocks™ Gene Fragments were mixed with the pLd2G backbone (gLTR_OUT) and genomic (gLTR_IN) amplicons and joined in a single reaction using In-Fusion^®^ HD Cloning Kit (Takara).

To construct plasmid for the Alpha.SIN.γRV.d2GFP.wPRE (AS.γRV) vector production, first, the EGFP in the original pAlpha.SIN.SF.EGFP.wPRE. plasmid [39] was exchanged for the d2GFP by digestion with NcoI and SpeI (NEB) restriction enzymes. The d2GFP and 3′ part of the proviral genome were amplified with pAS_GFP or pAS_PRE primers. The digested backbone and the two amplicons were joined by In-Fusion^®^ HD Cloning Kit (Takara). The SFFV U3 in the pAS.SFFV.d2GFP.PRE plasmid was then exchanged for other U3 sequences by digesting the plasmid with XcmI and AscI (NEB) restriction enzymes. The U3 parts were amplified by pAS_*γRV*U3_InFu primers, and the upstream and downstream U3-flanking sequences were amplified with pAS_XcmI, pAS_5intP, and pAS_AscI-intP primers. Resulting amplicons were mixed with the digested backbone and joined in a single reaction by In-Fusion^®^ HD Cloning Kit (Takara). All primers and oligonucleotide sequences are available in Appendix A.

### 2.3. Flow Cytometry

Cells were placed on a 96-well plate, spun down (1800 RPM, 3 min), and resuspended in PBS with Hoechst 33,258 (1000× diluted). Cells were analyzed using an LSRII (BD Biosciences, Franklin Lakes, NJ, USA) flow cytometer recording a maximum of 10,000 Hoechst-negative cells.

Before sorting, cells were spun down (10 min, 200× *g*) and resuspended in RPMI 1640 medium (Sigma). Cells were sorted with BD Influx (BD Biosciences) or FACSAria IIu (BD Biosciences) to cultivation medium and cultivated according to standard protocol.

### 2.4. Droplet Digital PCR (ddPCR)

Genomic DNA was purified by phenol-chloroform extraction and diluted with water to a concentration of around 10 ng/μL. A quantity of 1 μL of genomic DNA (10 ng) was added to a 20 μL reaction. Proviral copy number (CN) was quantified with vector-specific forward primer (q5GFP_MLV_F for MLV-derived vectors, q5GFP_pAS_F1 for AS vectors), q5GFP_R_2 primer, and FAM-labeled GFP-specific probe q5GFP_probe. RPP30 was used as a reference gene and quantified with RPP30_F, RPP30_R primers, and HEX-labeled probe RPP30_probe. ddPCR reaction was prepared with ddPCR™ Supermix for Probes (BioRad, Hercules, California, USA). Primers were added in the final concentration of 200 nM and probes with the concentration of 100 nM. Reactions were prepared in duplicates. Droplets were generated with 20 μL of reaction mix and 70 μL of Droplet Generation Oil for Probes (BioRad) with a QX200 Droplet Generator. Samples (40 μL) were transferred to 96-well plates. The amplification program was run as follows: 95 °C for 5 min; 40 cycles of 94 °C for 30 s, 58 °C for 30 s, and 72 °C for 30 s; and 98 °C for 10 min. The program was run with a ramp rate of 2 °C/s. QX200 droplet reader was used to read the samples. QuantaSoft Software version 1.7 (BioRad) was used to analyze raw data. CN of provirus and RPP30 was established as the mean of sample CopiesPer20uLWell values. Proviral per-cell CN was calculated as a CN of provirus/2x CN of RPP30.

### 2.5. DNA Library Preparation

The DNA library was prepared as described in [48] with slight modifications. Briefly, after phenol-chloroform DNA extraction from transduced cells, DNA was fragmented by DNA fragmentase (NEB, Ipswich, MA, USA) to obtain fragments 100–2000 bp long, with the main density between 500–1000 bp on an agarose gel. While purifying fragments on SPRI Magnetic beads (Canvax, Valladolid, Spain), we selected fragments 300–2000 bp long. Then, we repaired the ends and added 3′polyA overhangs with T4 polymerase, T4 polynucleotide kinase, and Taq DNA polymerase (all NEB). After purification with the High Pure PCR Cleanup Micro Kit (Roche, Basel, Switzerland), through T-A ligation, we ligated adaptors with Quick Ligase (NEB). Apart from the standard adaptor oligos Ion-Link-A and Ion-Link-Ba, we added blocking oligo Ion-Link-Bb to prevent adaptor–adaptor amplification. After another purification on columns, DNA was amplified with adaptor-specific and provirus-specific primers. In the PCR reaction mix, we added blocking oligos to prevent the amplification of inner proviral sequences. First, we ran a linear PCR reaction with biotinylated primer MLV-LTR1_F_Bio. After purification of the PCR reaction on the Dynabeads™ MyOne™ Streptavidin C1 beads (Invitrogen, Carlsbad, CA, USA), we ran a PCR reaction with the Ion_trP1 primer and the barcoded IonA_MIDx_mlvLTR2 primers. We isolated 300–500 bp long PCR products from an agarose gel. After purification on columns, the samples were quantified with the KAPA Library Quantification Kit for IonTorrent Platforms (Roche). The quantified samples were pooled in equimolar ratio and mixed with Ion Sphere Particles (Thermo Fisher Scientific, Waltham, MA, USA). The samples were sequenced on the IonTorrent platform (PGM sequencer; Thermo Fisher Scientific). The oligonucleotides are listed in Appendix A.

### 2.6. Integration Sites

First, reads in the FASTQ file were sorted according to the barcodes using the *cutadapt -g^B --overlap 8 --discard-untrimmed* command [49], where B stands for the barcode sequence and the read names were modified to contain the name of the barcode group. Reads containing a sequence of LTRs were sequentially selected first using cutadapt -g^GCTTGCCAAACCTACAGGTG --overlap 20 --discard-untrimmed command to select reads containing a sequence targeted by LTR-specific primer. Next, reads containing the last 9 nucleotides of the LTR sequence with residual sequence of the minimal length of 15 bp were selected using cutadapt -g^GGTCTTTCA --overlap 8 --discard-untrimmed --minimum-length 15 command. Adapter sequences were then removed from the reads with cutadapt -a ACCACTAGTGTCGAC --overlap 10 --minimum-length 15 command, and the reads containing amplified inner proviral sequence were removed using the cutadapt -g TTCCCCCCTT --overlap 10 --discard-trimmed command.

Trimmed FASTQ reads were mapped to both hg19 and hg38 human genome assemblies using the Bowtie 2 [50] with bowtie2 -p 20 -q -x hgX command, where the hgX marks the name of the assembly. Reads that map from the start of the read (“MD:Z:0” reads) that show a single hit in the genome (“XS:i:” reads) were selected and converted to BED file using samtools view -S -b [51] and bedtools bamtobed -cigar -i [52] commands. Each proviral integration site (IS) is recorded as a single genomic LTR-proximal position. ISs supported by at least 5 reads were selected. If more ISs were present within the distance of 5 bp, only the ISs supported by the most reads were selected. The two IS sets (IS_hg19.bed and IS_hg38.bed) were used independently against the annotations of respective genomic assemblies.

The set of random genomic ISs was created by the concatenation of three IS files and subsequent usage of bedtools shuffle command.

### 2.7. Distance to Features

We obtained the following annotations from publicly available databases: the genomic segments [53] from the University of California Santa Cruz (UCSC) genome annotation database (https://hgdownload.cse.ucsc.edu/goldenpath/hg19/database/wgEncodeAwgSegmentationChromhmmK562.txt, accessed on 11 May 2023); the chromatin subcompartments [54] from gene expression omnibus (SE63525_K562_Arrowhead_domainlist.txt.gz); and the lamina-associated domain (LAD) genomic coordinates from 4D Nucleome Data Portal [55,56] (data accession 4DNFIV776O7C). Distance to the features was calculated using bedtools closest -d command.

### 2.8. Software and Statistical Analysis

The cytometric data were gated using FlowJo™ v 10.10 Software (BD Life Sciences, Franklin Lakes, NJ, USA), and gated populations were exported to csv files. The exported data were subsequently concatenated and analyzed with custom R code (R Core Team 2022, Vienna, Austria). All statistical tests and plots were produced with R. Figures were produced with the ggplot2 package [57]. The “Impact” effect-size analysis was performed with the ImpactEffectsize package [58]. The schemes in the figures were created with BioRender (https://www.biorender.com/, Toronto, Canada).

## 3. Results

### 3.1. MLV LTR Drives Stable Transgene Expression of Bin Vectors

In our previous work, we showed that cells transduced by MLV-derived vectors express the transgene stably for two months [5]. Also, Van Looveren et al. (2021) showed that promoters derived from spleen focus-forming virus (SFFV) long terminal repeat (LTR) and the eukaryotic translation elongation factor 1 α (EF1α) establish long-term stable expression of the transgene in the context of Bin vectors with altered integration preference. Here, we adapted a parallel approach to test whether the disruption of the natural promoter/enhancer preference of MLV integration also disrupts the ability of MLV LTRs to drive high, long-term stable expression. We utilized a previously used MLV-derived mini-vector LG [43] and produced the vectors with wt integrase (IN^wt^) or Bin vectors with W390A mutation (Bin^W390A^) or with fused CBX chromodomain (Bin^CBX^) (Figure 1).

First, we transduced the K562 cell line, a widely used leukemia cell line with well-characterized epigenome, and measured the GFP expression at three days post infection (3 dpi) by flow cytometry. All vectors efficiently expressed GFP with no marked differences in the intensity of expression (Figure 1 and Appendix A). Next, we followed the long-term stability of expression in the population of transduced cells. We observed that the provirus-expressing cells form a stable part of the population for at least 30 dpi, irrespective of the vector used or the multiplicity of infection (Figure 1 and Appendix A). We also followed proviral expression in the polyclonal GFP+ populations bulk-sorted at 3 dpi (Figure 1). The GFP+ cells formed a stable fraction of the cell population. At 23 dpi, we observed the lowest GFP+ proportion in INwt-vector-transduced cells. However, the GFP+ cells still formed more than 75% of the population, and this fraction has been observed unchanged since 14 dpi.

Finally, we examined the expression in single-cell clones expanded from the low-multiplicity transduced single GFP+ cells sorted at 3 dpi. We characterized the expression of 200 single-cell clones in each vector-defined group at 30 dpi and observed that in all three groups, the majority of the clones contained more than 90% of GFP+ cells (Figure 1). The intensity of GFP expression was not significantly changed in the clonal populations transduced with Bin vectors (Figure 1). Some expression instability was observed only among the clones transduced by the IN^wt^ vector, where 11 clones (5.5%) showed less than 90% of GFP+ cells, from which 2 clones contained even less than 50% of GFP+ cells. No clones with less than 90% GFP+ cells were observed among those transduced with Bin vectors. The data thus suggest that the MLV proviruses that are active at 3 dpi are resistant to silencing of gene expression for at least one month, irrespective of the integrase variant used.

### 3.2. Expressed MLV Proviruses Show a Retargeted Integration Site Profile

The Bin vectors were reported to integrate less effectively into traditional MLV sites marked by open chromatin histone modifications with a shift toward silent heterochromatin [28]. This shift in the integration site (IS) distribution of Bin vectors was observed in both transcriptionally active and non-selected proviral populations [29]. Since we observed no differences in the expression stability between MLV IN^wt^ and Bin vectors, we investigated whether the expected differences in IS distribution are preserved in the populations of expressed proviruses. We sorted the polyclonal populations of GFP+ K562 cells at 3 dpi and analyzed the distribution of proviral ISs according to known genomic and epigenomic features.

As the Bin^CBX^ vector was reported to display the most significantly retargeted profile, we tested whether there is any significant difference in proviral IS distribution between the active proviruses of IN^wt^ and Bin^CBX^ vectors. First, we calculated the distances to the nearest genome segment [53] for each IS and compared the distance distributions between the vector groups. We used the Impact” effect-size analysis [58] to quantify the segment-relative distance differences between the distribution of IN^wt^ and Bin^CBX^ ISs. While there is a low Impact observed for the majority of sequences (68% with absolute value of Impact below 0.3), there are several segments with medium-to-high Impact (above 0.5) (Figure 2 and Appendix A). High Impact was associated with central tendency difference (CTDiff) as well as the difference in distribution shape (Appendix A) of proviral distance distribution relative to the genome segments associated with the active transcriptional start sites (Tss, TssF), enhancers (Enh, EnhF), and the segment associated with the 5′ end of transcription units (Gen5) (Figure 2). The common characteristics of those segment-relative distance distributions is that the median distance to the features is very low and is gradually increased in both Bin vector ISs, with the Bin^CBX^ vector ISs displaying the highest median distance to the segments (Figure 2 and Appendix A). For instance, the median distance to the Tss segment was 0.8 kb, 11.4 kb, and 22.6 kb in the IN^wt^, Bin^W390A^, and Bin^CBX^ IS data sets. The median IS distance to the Enh segment increased from 1.8 kb in the IN^wt^ data set to 8.5 kb and 15.6 kb in the Bin^W390A^ and Bin^CBX^ IS data sets. On the other hand, there were no or very marginal changes in distribution toward the segments associated with weak enhancers (EnhW and EnhWF) and DNase hypersensitive sites (DNaseD, DNaseU, and FaireW).

Next, we tested whether the active proviruses showed altered distribution in chromatin domains. We collected the coordinates of chromatin A/B subcompartments and lamina-associated domains (LADs) and analyzed the frequency of proviruses in those domains. The majority of proviruses were found in the A subcompartments that are composed of transcriptionally active chromatin (Figure 2). Specifically, 68% and 17% of the IN^wt^ ISs were found in the A1 and A2 subcompartments. While the Bin^W390A^ ISs resembled the IN^wt^ ISs in A subcompartment targeting, the Bin^CBX^ ISs were found within the A1 subcompartment with decreased frequency (56%) and were increased in the A2 subcompartments (21%). Despite this minor shift, the Bin^CBX^ ISs were still significantly enriched in the A1 subcompartments compared to the random control sites (Figure 2).

In contrast to the A compartments, the B compartments and LADs are mostly composed of transcriptionally repressed chromatin and represent the infrequent locations of the active MLV proviruses. The B1, B2, and B3 subcompartments and LADs contained 8%, 1%, 2.2%, and 2.6% of the IN^wt^ ISs. Similarly to the A subcompartments, the Bin^W390A^ ISs showed only a moderate change compared to the IN^wt^ IS distribution. However, the Bin^CBX^ ISs were enriched more than 2-fold in the B2 and B3 subcompartments and LADs (Figure 2). The highest enrichment was observed in the LADs, where almost 9% of active Bin^CBX^ ISs were found (Figure 2). Despite this increase, this frequency was still far below the expected 50% produced by random genome targeting (Figure 2).

Here, we showed that the stably active MLV proviruses of Bin vectors display an altered profile of IS distribution. Compared to the proviruses of the IN^wt^ vectors, the active MLV proviruses of Bin vectors were positioned further away from the TSS and enhancers and more frequently situated in the heterochromatin-associated compartments.

### 3.3. The Stable Expression after Integration Retargeting Is General for Gammaretroviruses

The previous results suggest that MLV-derived vectors can establish and maintain stable expression in a non-preferred epi/genomic environment. We further asked if this feature is common to other γRVs or if it is specific to MLV. We constructed gammaretroviral mini-vectors where the GFP with fused destabilization domain (d2GFP) transcription is controlled by the LTRs derived from Moloney murine leukemia virus (MoMLV), feline leukemia virus (FeLV), spleen necrosis virus (SNV), koala retrovirus (KoRV), and cervid endogenous retrovirus (CrERV) (Figure 3). All constructs are capable of LTR-driven expression of d2GFP after transfection (Appendix A). Except for the MoMLV LTR that displays a lower intensity of post-transfection expression, other LTRs drive the d2GFP expression comparable to the CMV promoter. We thus produced the vectors for the d2GFP transduction with the MLV-based packaging system utilizing the IN^wt^, Bin^W390A^, and Bin^CBX^ variants and evaluated expression stability in the transduced K562 cell line.

At 3 dpi, we observed vector expression in the transduced cells of each vector group (Figure 3). The FeLV- and SNV-derived vectors expressed GFP to an extent similar to the MoMLV-derived vector. On the other hand, transduction by the KoRV- and CrERV-derived vectors was less efficient, transduction by the KoRV-derived vector produced a low number of low-intensity vector-expressing cells, and the CrERV-derived vector expression was dim and close to the intensity of the negative population. When cells were transduced with the IN^wt^ vectors, the FeLV-vector-transduced cells displayed a 1.8- and 1.5-fold lower median of GFP intensity than the MoMLV- and SNV-transduced cells, respectively. Although the transduction efficiency does not seem to be altered in the Bin-vector-transduced cells, we observed slight shifts in the expression intensity in the GFP+ cells. We observed a decrease in the median GFP fluorescence intensity in the MoMLV- and SNV-derived Bin^CBX^-vector-transduced cells compared to the IN^wt^ variant (1.2- and 1.5-fold, respectively, Appendix A). In contrast, using the FeLV-derived vector, we observed a 1.4-fold increase in expression intensity in cells transduced by the Bin^CBX^ variant compared to the IN^wt^ variant. In fact, the median expression intensity of the FeLV Bin^CBX^ vector is identical to the SNV IN^wt^ vector and only 1.2-fold lower than the MoMLV IN^wt^-vector-transduced cells. Although the shifts in the expression intensities are significant, they do not represent serious changes in the expression patterns of transduced genomes.

Next, we followed expression in the transduced cells for two weeks post infection (wpi) (Figure 3 and Appendix A). The FeLV-derived vector displayed expression stability comparable to the stability observed with MoMLV. On the other hand, the SNV-derived vectors displayed a moderate decrease in the number of effectively transduced cells. The KoRV-derived vector-transduced cells displayed high variability of expression stability, which can be attributed to the very low number of effectively transduced cells at 3 dpi. Unlike other vector-transduced cell groups, the CrERV-transduced cells disappeared soon after the initial measurement at 3 dpi. It is noteworthy that the Bin vectors could be traced over two weeks without any significant decrease in the percentage of effectively transduced cells compared to the IN^wt^ variant. We also observed no significant changes in the vector expression intensity over time (Appendix A). Thus, although there might be slight differences between the vectors, the expression profiles were stable during the course of four independent experiments.

Finally, we quantified the per-genome copy number (CN) of GFP in the transduced polyclonal populations at 2 wpi. We then normalized the percentage of GFP+ cells in the population to the genome CN to estimate a fraction of expressed proviruses (Figure 3 and Appendix A). In all samples, the ratio was below 0.8, suggesting that a population of non-expressed vectors is present in each vector-transduced population. The ratio was highest in the MoMLV-IN^wt^-transduced cells and slightly decreased to 0.6 in the MoMLV Bin^W390A^-, MoMLV Bin^CBX^-, and all FeLV-transduced cells. In the SNV-transduced cells, the ratio ranged between 0.3 and 0.5, which would suggest that more than half of integrated vectors with the SNV LTRs are silent at 2 wpi. Similar ratios were observed in the KoRV-derived vector-transduced cells, although due to a low number of GFP+ cells, there are high deviations between data from different experiments. Generally, low CNs of vector genomes were observed in the CrERV-derived vector-transduced cells, suggesting that the loss of actively transduced cells is possibly due to the defect in integration but not due to expression silencing.

Here, we showed that the γRV mini-vectors equipped with FeLV, SNV, and KoRV LTRs can be produced with the MLV-derived packaging system and that they can be used for stable transduction of human cells. The usage of an alternative Bin packaging system did not abrogate the stability of transgene expression in any of the vector systems. The FeLV- and SNV-derived vectors proved to be capable of expressing GFP with high intensity, while the KoRV-derived vectors produced only a few low-expression transduced cells. The FeLV-derived Bin^CBX^ vector is particularly noteworthy for its efficient vector production and transduction efficiency, which is comparable to the MLV-derived vectors.

### 3.4. Gammaretroviral LTRs as Internal Promoters in αRV SIN Vector

Previously, we studied the activity of the γRV LTRs in the context of γRV minigenome vectors derived from the MLV packaging system and demonstrated effective cell transduction with MoMLV, FeLV, and SNV-derived vectors. To further investigate the activity of γRV LTRs in different genomic contexts, we examined the promoter activity of LTRs in ASLV-derived vectors that display no significant preference for any specific genomic feature [8,11]. For this purpose, we inserted the 3′ unique (U3) parts of the MoMLV, FeLV, SNV, CrERV, or KoRV LTRs as internal promoters in the ASLV-derived SIN (AS) vector originally equipped with the SFFV-derived U3 segment acting as an internal promoter/enhancer (Figure 4). We then compared the post-transduction expression stability of the resulting AS.γRV.d2GFP vectors in the K562 cell line.

All AS.γRV.d2GFP vectors effectively transduced K562 cells, producing up to 40% GFP+ cells at 3 dpi (Appendix A). Analysis of the expression intensities of the GFP+ cells revealed significant differences between the vectors (Figure 4, Appendix A). Cells transduced with AS.MoMLV, AS.SFFV, and AS.FeLV vectors reached the highest intensities among the AS.γRV vectors. The median intensity of GFP expression was about 2-fold lower in cells transduced with AS.SNV and AS.CrERV vectors (Appendix A). The AS.KoRV-transduced cells were present with the lowest expression intensities, 5-fold lower than the AS.MoMLV-vector-transduced cells and 2-fold lower than the AS.SNV-vector-transduced cells.

Next, we examined stability of expression as the proportion of GFP+ cells in the transduced populations during a long-term culture (Figure 4). The proportion of GFP+ cells in AS.MoMLV, AS.SFFV, and AS.FeLV-transduced populations first slightly increased after 3 dpi and stayed stable after. Also, AS.KoRV-transduced cells showed stable levels of GFP+ cell proportion. The proportion of GFP+ cells in the AS.CrERV-transduced population was stable for about two weeks after transduction, but then started to slowly decrease at 17 dpi and reached about 83.5% (±1.6%) at 31 dpi, the proportions observed at 3 dpi. We observed a rapid decrease in the proportion of GFP+ cells in the AS.SNV-transduced cells (Appendix A). At 31 dpi, the mean GFP+ cell proportion reached 37.5% (±5.6%) of the proportions observed at 3 dpi.

We collected genomic DNA from the low-multiplicity transduced populations at 14 dpi and performed ddPCR to estimate the proviral CN in each vector-transduced population. At 14 dpi, the proportion of the GFP+ cells was comparable in all selected populations (mean 3.9% ± 0.6%). The proviral CN, however, fluctuated between 3.9 (AS.SFFV) and 8.8 (AS.CrERV) proviral copies per 100 equivalents of cellular genomes (Figure 4). We then estimated the active genomes ratio and observed that AS.SFFV together with AS.FeLV are the most active vectors, producing between 80 and 90% of active proviruses at 14 dpi. AS.MoMLV and AS.KoRV displayed lower proportions of active genomes between 60 and 70%, the ratio observed with Bin vectors. We observed the lowest proportion of active genomes (36%) in AS.SNV-transduced cells. As the overall proportion of GFP+ cells decreases over time, we estimate that already at 3 dpi, the proportion of active genomes of AS.SNV is 46% and may go down to 13% at 31 dpi (Appendix A).

Here, we showed that γRV LTRs can be used as internal promoters in SIN vectors. In the context of αRV SIN vector, MoMLV and FeLV LTRs produce high-intensity and stable expression of the vector at levels comparable to the SFFV LTR. However, contrary to MoMLV LTR, FeLV LTR produces a low frequency of silenced proviruses early after integration. KoRV LTR also produced a long-term stable proportion of vector-expressing cells, albeit with much lower expression intensity than other vectors.

## 4. Discussion

The genomic and epigenetic environment at the site of proviral integration is one of the key determinants of proviral expression. Because γRVs exhibit a strong preference for integration into active promoters and enhancers, regions associated with protection of retroviral transcription [3,4,6,20], we investigated whether γRV LTR promoter activity is dependent on the environment that γRV integration prefers. Our results demonstrate that despite a significant proportion of proviruses silenced early after integration, distinct γRV LTRs establish and maintain long-term stable expression, even after the disruption of the natural integration preference.

First, we quantified the post-integration expression stability of γRV LTRs. We observed only limited silencing or variegation of γRV LTR-driven gene expression at 3 dpi, not only with Bin vectors (Figure 3) but also with γRV LTRs as internal promoters in the αRV SIN vector (Figure 4). This observation is consistent with previous reports using the IN^wt^ MLV-derived vector [5] or Bin vectors with heterologous internal promoters [29]. We observed strong silencing after 3 dpi only with SNV LTR as an internal promoter in the αRV SIN vector (Figure 4). The RU5 part of the SNV LTR that is missing in the internal SNV promoter was shown to act as a positive post-transcriptional control element enhancing cytoplasmic expression of RNA [59,60,61]. Our results suggest that the RU5 region may also play a positive role in transcription stability. Although rarely observed, the variegation of expression is most frequent in the MLV vector with natural integration preference (Figure 3). Notably, the variegation of MLV expression is observed only when the expression of individual proviruses is tracked in clonal populations (Figure 1). The presence of variegating MLV proviruses thus seems to depend on integration site selection, although, unexpectedly, variegation is observed in the population of proviruses with a natural preference for promoters and enhancers.

Next, we observed that with normal integration targeting in the K562 cell line, about 20% of MLV proviruses are silent early after infection (Figure 3), i.e., before the markers of proviral expression are first detected (3 dpi in our study). The silent population can consist of both epigenetically silenced and defective proviruses [30,31]. Our observation that the proportion of silent proviruses doubles after integration retargeting with Bin vectors (Figure 3) indicates the epigenetic mechanisms of silencing and further supports the concept that promoter activity of MLV LTR is, at least to some extent, sensitive to the epigenetic environment at the target site. Missing expression variegation and the increase in the silent population of MLV proviruses after retargeting suggest that BET-directed integration targets MLV to specific, semi-permissive sites missed by Bin vectors. The phenomenon of an increased proportion of the silenced population after integration retargeting seems to be specific to MLV LTR as we observed a uniform proportion of the silent population among wt and Bin vectors bearing FeLV and SNV LTRs. One explanation could be that, unlike MLV LTRs, FeLV and SNV LTRs are rapidly silenced at putative semi-permissive sites. Our results show that significant transcriptional silencing of all tested γRV LTRs occurs rapidly after proviral integration. The proviruses that escape the early silencing events then perform with long-term stable expression. The early post-integration silencing of γRV LTRs in somatic cells is a phenomenon that should be addressed in future studies.

We showed that the natural integration preference of γRV directed by BET-integrase interaction is dispensable for the establishment of long-term stable proviral expression. On the other hand, although retargeted proviruses are more distant from promoters and enhancers, most of the retargeted active proviruses are still located within predicted chromatin A compartments and thus can be supported by the transcriptionally permissive chromatin environment (Figure 2). We also cannot rule out the contribution of selection to the observed profile of retargeted integration. However, we observed significant differences between active proviruses of IN^wt^, Bin^W390A^, and Bin^CBX^ vectors, and there were only marginal differences between non-selected and active proviral populations of Bin vectors bearing an internal SFFV promoter [29]. While these observations do not contradict the IS-specific silencing, the effect of non-permissive sites on tested γRVs LTRs is probably too weak to be detected by current bulk analysis techniques. Some stochastic shutdown of proviral expression immediately after integration can also occur.

In conclusion, we show that diverse γRV LTRs can function as promoters in diverse epigenetic environments outside active promoters and enhancers. This work opens the possibility of studying other than MLV γRVs as drivers of cassette expression.

## Figures and Tables

**Figure 1 viruses-16-01518-f001:**
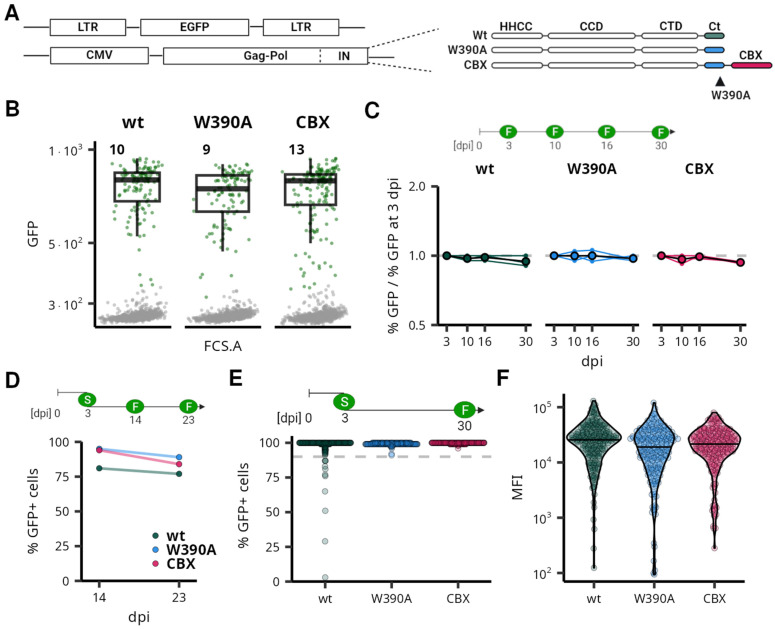
MLV is stably expressed after integration retargeting. Comparison of expression stability of MLV-derived vectors after integration retargeting. (**A**) A schematic depiction of the LTR-dGFP-LTR vectors and integrase (IN) variants used in the experiment. (**B**) A dot plot representing the flow cytometry measurement of K562 cells transduced by MLV-derived vectors 3 dpi. Numbers correspond to the percentage of GFP+ cells. For each vector variant, 2400 live cells were selected to construct the dot plot. (**C**) Fold change in the fraction of GFP+ cells in the transduced population during 30 days of culture. The *y*-axis is on a log_2_ scale. Timepoint 3 dpi represents the data in panel (**B**). (**D**) Fraction of GFP+ cells after the cultivation of bulk populations sorted for GFP expression at 3 dpi. Panels (**E**,**F**) demonstrate characteristics of clonal populations expanded from single cells sorted for GFP expression at 3 dpi. The fraction of cells expressing GFP (**E**) and the mean fluorescence intensity (MFI) (**F**) were measured at 30 dpi. In each category, 201 clonal populations were characterized. Schemes in panels (**C**–**E**) show a time course of experiments, with flow cytometry (F) or FACS sorting (S) performed at a given dpi.

**Figure 2 viruses-16-01518-f002:**
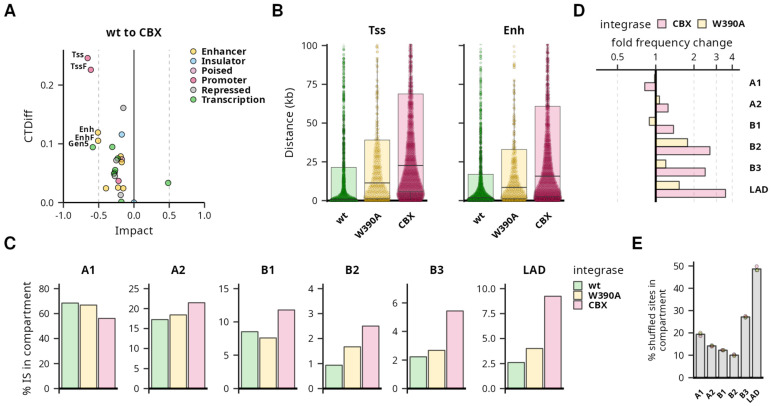
Integration site (IS) profile of active MLV proviruses after retargeting. IS distribution analysis of MLV IN^wt^ and Bin vectors in K562 cells expressing GFP. Panels (**A**,**B**) represent an analysis of IS distances to defined chromatin segments. (**A**) Dot plot representing the results of the “Impact” effect-size analysis comparing IS of Bin^CBX^ and IN^wt^ vector usage on the distribution of ISs. Points represent individual segments grouped into categories differentiated by colors. CTDiff marks the change in the central tendency of the distribution. Shown are the names of the segments with Impact absolute value ≥ 0.5. (**B**) Plot showing the distribution of IS distances to the active transcription start site (Tss) and strong enhancer (Enh) chromatin segments. Each dot represents an individual IS, box plots represent medians and quartile range of the distance distributions. (**C**–**E**) Panels represent the targeting of chromatin A/B subcompartments and lamina-associated domains (LAD). (**C**) A bar plot representing a fraction of proviruses integrated into the subcompartments and LADs. Fold frequency changes in subcompartment targeting of W390A and CBX IN variants to wt IN. The *x*-axis is depicted in the log_2_ scale. (**E**) Frequency of shuffled sites in the chromatin subcompartments representing a random targeting control. Each dot represents a shuffled site set prepared for each of the IN variant samples. The height of the bar represents the mean targeting frequency.

**Figure 3 viruses-16-01518-f003:**
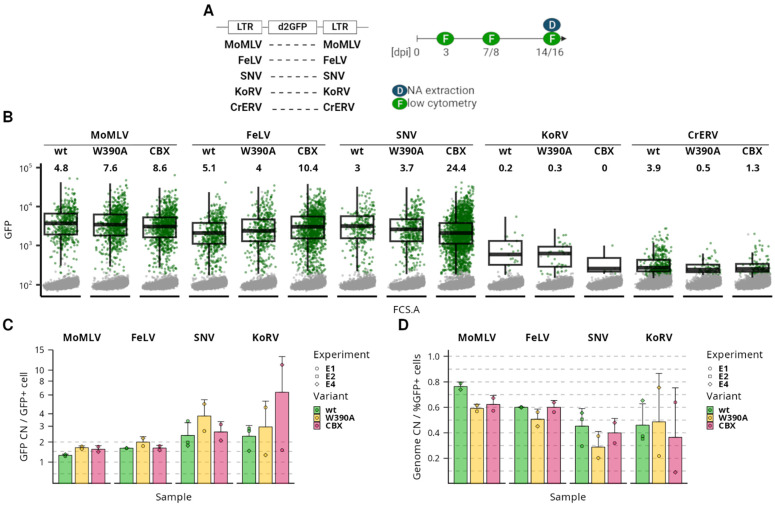
Expression stability of non-MLV gammaretroviral vectors after retargeting. (**A**) Gammaretroviral vectors carrying LTRs from five different retroviruses were constructed. Viral stocks were produced with Gag-Pol variants, and expression of the proviruses in transduced cell population was observed for two weeks. (**B**) Expression of d2GFP by gammaretroviral vector-transduced cells at 3 dpi. For each sample, 10,000 cells are shown. GFP-positive cells are displayed in green. Box plots show the median and quartile range of GFP intensity for the GFP-positive population. Numbers specify the percentage of GFP-positive cells in the transduced population. Shown data represent experiment E4. Log_10_ transformed GFP and FSC.A signal is used in a graph. (**C**) Bar plot showing the change in % of GFP-positive cells in time. The values are relative to the level of expression observed at 3 dpi of a particular experiment. (**D**) Ratio of % GFP-positive cells to a copy number (CN) of detected d2GFP-encoding genomes per 100 cells. The flow cytometry and DNA extraction were performed two weeks after transduction.

**Figure 4 viruses-16-01518-f004:**
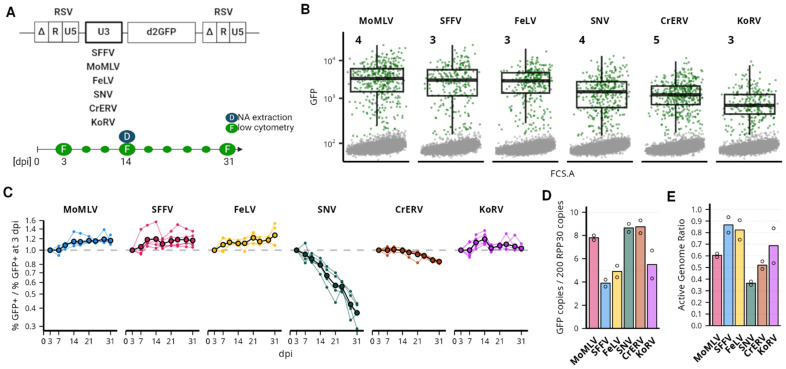
Intensity and stability of αRV vector expression with γRV LTR as an internal promoter. (**A**) A schematic depiction of an αRV AS vector. The internal promoter was derived from the U3 promoter/enhancer part of the LTR of the studied γRVs. The bottom of the panel contains the scheme of the experiment, where GFP expression was followed every 3–4 days at 3–31 dpi. (**B**) Intensity of d2GFP expression in transduced K562 cells at 3 dpi. Cells inside the GFP+ gate are colored green. Box plot describes the intensity distribution of GFP+ cells. The numbers show the percentage of GFP+ cells of all alive cells. For each sample, 10,000 cells are shown. Samples are ordered by the median intensity of GFP+ cells. (**C**) Representation of the time-course experiment where the percentage of GFP+ cells in transduced populations was followed. Values on the *y*-axis show the percentage of GFP+ cells relative to the percentage of GFP+ cells observed at 3 dpi. Light lines and points show individual transduction experiments with divergent multiplicities of infection. Black-outlined points connected by black lines show the average of all experiments. (**D**) Copy number of proviruses (GFP copies) as per 100 genomic equivalents (200 copies of RPP30 reference target) measured by the droplet digital PCR (ddPCR). Proviral copy number was established from genomic DNA collected at 14 dpi in samples shown in panel (**B**). (**E**) Ratio of the percentage of GFP+ cells per proviral copy number per 100 genome equivalents. The value of 1 marks the point where all proviruses are expected to be active in expression. Points in (**D**,**E**) show values of technical duplicates.

## Data Availability

Raw sequencing data and integration site coordinates are available from Gene Expression Omnibus under the accession number GSE269015. The code is available at https://github.com/dalibormiklik/gammaLTR_activity.git.

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
