# Peer review of "Long Terminal Repeats of Gammaretroviruses Retain Stable Expression after Integration Retargeting"

_viruses, 2024, doi:10.3390/v16101518_

Round 1
Reviewer 1 Report
Comments and Suggestions for Authors
This manuscript presents interesting results on the properties of gammaretroviral vectors whose integrase proteins have been modified so that their integration site preferences have been altered. In general, these changes in targeting have little effect on the levels or efficiency of expression from the vectors.
My only suggestion is that the presentation could be a little clearer. In many of the figures, the labeling is much too small for a senior reader like myself. The text also sometimes leaves too much to the intelligence of the reader. For example, lines 357-359 state that “active proviruses…are further away from TSS and enhancers and more frequently situated…” “Further away/more frequently situated” than what?? Finally, the Discussion summarizes the results, which is extremely helpful, but would be even more helpful if each statement of conclusions were accompanied by identification of the specific figure leading to the conclusion.
Should the pH of the HBS in the transfections be indicated?
Line 439: I believe “any” would be more appropriate than “none”.
Author Response
We want to thank the reviewer for spending the time to carefully read our manuscript. We went through the manuscript and made modifications according to the reviewer's suggestions.
Comments 1: In many of the figures, the labeling is much too small for a senior reader like myself.
Response 1: We modified all Figures 1, 2, 3, and 4, so the labels in the figures are larger and easier to read.
Comments 2: The text also sometimes leaves too much to the intelligence of the reader. For example, lines 357-359 state that “active proviruses…are further away from TSS and enhancers and more frequently situated…” “Further away/more frequently situated” than what??
Response 2: We modified the text of the manuscript in several places to improve the readability of the text. The mentioned sentence was modified: "Compared to the proviruses of the INwt vectors, the active MLV proviruses of Bin vectors were positioned further away from the TSS and enhancers and more frequently situated in the heterochromatin-associated compartments."
Comments 3: Finally, the Discussion summarizes the results, which is extremely helpful, but would be even more helpful if each statement of conclusions were accompanied by identification of the specific figure leading to the conclusion.
Response 3: We modified the text in the Discussion section. The Discussion now contains references to figures supporting the statements made in the text.
Comments 4: Should the pH of the HBS in the transfections be indicated?
Response 4: We added the information about the pH of the HBS used in the transfection: "HBS (pH 7.0-7.1)".
Comments 5: Line 439: I believe “any” would be more appropriate than “none”.
Response 5: We corrected the sentence at line 439. The new sentence is: "The usage of an alternative Bin packaging system did not abrogate the stability of transgene expression in any of the vector systems.".
Reviewer 2 Report
Comments and Suggestions for Authors
This paper reports a careful comparison of several gammaretroviral LTR promoters in driving GFP expression from proviruses, when targeted either by wt MLV IN (via Bet interactions) or by mutant INs. Both early and late expression are examined. Both bulk and individual clones were examined (an impressive set of experiments). The integration profiles were also examined, and described in terms of distance from known features of the genome. Tests with the promoters embedded in ASV vectors were also performed, where integration is not specifically targeted.
The results are clearly presented in excellent figure formats and clearly described in text. The magnitudes of the effects are not overstated and when less than two-fold, as often seen, are stated simply and accurately. In most cases with the MLV LTR, expression is high and stable, independent of integration targeting. Other LTRs gave interesting differences, with some being generally weaker, and some much less stable over time (e.g. SNV).
All told, I found this to be a clear, balanced and compelling examination of a variety of interesting variables – LTRs, and integration site preference – on expression.
I found no significant problems and support acceptance.
Small points:
Line 439: I believe the sentence written as “did not abrogate the stability of transgene expression in none of the vector systems” should be “in any of the vector systems”
In several places, sentences are begun with “Noteworthy,”….this is a grammar problem. “Noteworthy” is an adjective and shouldn’t just stand alone. Perhaps “It is noteworthy that…” would be better.
Author Response
We want to thank the reviewer for spending the time reading our manuscript and for the encouraging comments. We went through the manuscript and made modifications according to the reviewer's suggestions.
Comments 1: Line 439: I believe the sentence written as “did not abrogate the stability of transgene expression in none of the vector systems” should be “in any of the vector systems”
Response 1: We corrected the sentence at line 439. The new sentence is: "The usage of an alternative Bin packaging system did not abrogate the stability of transgene expression in any of the vector systems.".
Comments 2: In several places, sentences are begun with “Noteworthy,”….this is a grammar problem. “Noteworthy” is an adjective and shouldn’t just stand alone. Perhaps “It is noteworthy that…” would be better.
Response 2: We corrected the grammar in several places of the text. This includes sentences on Line 416 (“It is noteworthy that the Bin vectors could be traced over two weeks without any significant decrease in the percentage of effectively transduced cells compared to the INwt variant.”) and 442 (“The FeLV-derived BinCBX vector is particularly noteworthy for its efficient vector production and transduction efficiency which is comparable to the MLV-derived vectors.”) of the revised version of the manuscript.